# Direct determination of high-order transverse ligand field parameters via μSQUID-EPR in a Et$_4$N[$^{160}$GdPc$_2$] SMM

Gheorghe Taran[1], Eufemio Moreno-Pineda [2,3] ✉, Michael Schulze [1], Edgar Bonet[4], Mario Ruben [5,6,7] ✉ & Wolfgang Wernsdorfer [1,7] ✉

The development of quantum technologies requires a thorough understanding of systems possessing quantum effects that can ultimately be manipulated. In the field of molecular magnetism, one of the main challenges is to measure high-order ligand field parameters, which play an essential role in the relaxation properties of SMMs. The development of highly advanced theoretical calculations has allowed the ab-initio determination of such parameters; however, currently, there is a lack of quantitative assessment of how good the ab-initio parameters are. In our quest for technologies that can allow the extraction of such elusive parameters, we develop an experimental technique that combines the EPR spectroscopy and μSQUID magnetometry. We demonstrate the power of the technique by performing EPR-μSQUID measurement of a magnetically diluted single crystal of Et$_4$N[GdPc$_2$], by sweeping the magnetic field and applying a range of multifrequency microwave pulses. As a result, we were able to directly determine the high-order ligand field parameters of the system, enabling us to test theoretical predictions made by state-of-the-art ab-initio methods.

Our ability to develop new technologies is greatly enhanced by leveraging the unique properties of quantum systems. By gaining control over the quantum states of the system, it is possible to achieve unparalleled performance in a wide range of fields. For instance, information storage in high-density devices permits to store an unprecedented amount of data in increasingly smaller physical units. Furthermore, the use of quantum states in computation can potentially outperform current state-of-the-art supercomputers. The ability to exploit the coherent superposition and interference of quantum states opens the door to new algorithms that cannot be implemented using traditional computing methods[1–3]. The control of information at

a quantum level promises to revolutionise how information is processed and stored[4,5].

Single-Molecule Magnets (SMMs), molecules with an energy barrier to the relaxation, have shown bewildering quantum effects ranging from Quantum Tunnelling of the Magnetisation (QTM)[6,7], quantum coherence[8,9], Berry phases[10], quantum oscillation[11,12] and entanglement[13,14]. The quantum nature of these systems prompted their proposal for technological applications such as high-density data storage devices[15–17], quantum computers[18–20], quantum sensing[21–23] and quantum communication[24,25]. The possibility of chemically modifying the periphery, as well as the properties of the

[1]Physikalisches Institut, Karlsruhe Institute of Technology, D-76131 Karlsruhe, Germany. [2]Depto. de Química-Física, Facultad de Ciencias Naturales, Exactas y Tecnología, Universidad de Panamá, Panamá, Panamá. [3]Grupo de Investigación de Materiales, Facultad de Ciencias Naturales, Exactas y Tecnología, Universidad de Panamá, Panamá, Panamá. [4]Univ. Grenoble Alpes, CNRS, Grenoble INP, Institut Néel, Grenoble 38000, France. [5]Centre Européen de Sciences Quantiques (CESQ) within the Institut de Science et d'Ingénierie Supramoléculaires (ISIS), 8 allée Gaspard Monge, BP 70028, 67083 Strasbourg Cedex, France. [6]Institute of Nanotechnology (INT), Karlsruhe Institute of Technology (KIT), Hermann-von-Helmholtz-Plats 1, D-76344 Eggenstein-Leopoldshafen, Germany. [7]Institute for Quantum Materials and Technology (IQMT), Karlsruhe Institute of Technology (KIT), Hermann-von-Helmholtz-Platz 1, D-76344 Eggenstein-Leopoldshafen, Germany. ✉e-mail: eufemio.moreno@up.ac.pa; mario.ruben@kit.edu; wolfgang.wernsdorfer@kit.edu

metallic ion(s) embedded in the molecule, makes these systems very appealing for many applications. Chemical synthesis, furthermore, allows for the creation of precise and reproducible systems, an important step towards scalability. The control over the synthesis of SMMs has been demonstrated by the rational chemical design of SMMs that could potentially store information above liquid nitrogen temperatures[16,17]. Furthermore, SMMs can act as scaffolds for quantum information processing (QIP), leading to long coherence times and the possibility to manipulate their states to implement quantum algorithms[18–20,26]. Likewise, their multilevel character can be employed as so-called qudits, systems with an increased number of available states for computation[18,27,28], as demonstrated with photons[29] and molecules[26,30]. Expansion of the number of electronic or nuclear states of qudits, is likewise achievable by coupling the electronic states[31–33] and by employing an isotopologue approach[34–36].

SMMs are appealing quantum objects for advanced applications. Nonetheless, for their successful integration into quantum technologies, the parameters governing their magnetic anisotropy must be unravelled[37,38]. This task is far from trivial –determining the anisotropy parameters in SMMs is extremely difficult, especially for complexes with low symmetry[38]. Two important experimental techniques that were previously used to study the ligand field of SMMs are Electron Paramagnetic Resonance (EPR)[39] and Super Conducting QUantum Interference Devices (SQUIDs)[40]. EPR spectroscopy is a rather versatile technique that allows the investigation of the electronic properties of SMMs[41]. For example, an on-chip investigation of $^{155,157}$Gd ions doped in CaWO$_4$ showed the possibility to control the electronic and nuclear degrees of freedom[42]. Moreover, EPR has been coupled to very advanced techniques, such as Scanning Tunnelling Microscopes (STMs) ultimately leading to single atoms EPR spectroscopy[43–48]. On the other hand, μSQUID arrays are some of the most sensitive detection systems and were extensively used in the study of both dynamic and static properties of SMMs[40]. For example, the observation of QTM in TbPc$_2$ through μSQUID measurements[6,7] later enabled the implementation of Quantum Grover's algorithm at a single-molecule level[26]. Although EPR and μSQUIDs are two of the most sensitive and important techniques for the comprehension of SMMs, these techniques offer limited sensitivity for the determination of high-order ligand field parameters in bulk, which play a major role in the relaxation characteristic of SMMs. The deployment of highly advanced Complete Active Space Self-Consistent Field Calculations (CASSCF) is hence the most popular, and often, the only method that allows the determination of such important parameters[38].

In this work, we showcase the merge of the EPR and μSQUID techniques to study the magnetic anisotropy of SMMs. The technique consists of the application of microwave multifrequency pulses during the μSQUID loops acquisition, hence, simultaneously perturbing and measuring the electronic population of the different energy levels of the spin system. As a probe of principle, we study an Et$_4$N[$^{160}$GdPc$_2$] isotopologue SMM, which features uniaxial magnetic anisotropy at sub-Kelvin temperatures. Through the 3D control of the applied magnetic field and access to multifrequency RF pulses, we construct a high-resolution map of the energy spectrum of the system that in turn allows for the precise determination of the higher-order ligand field parameters.

## Results

### Et$_4$N[GdPc$_2$] single ion magnet (SIM) and spin hamiltonian

To test the technique, we study a gadolinium-based double-decker complex (Fig. 1a). Although typically thought as isotropic, Gd$^{3+}$ complexes can exhibit anisotropy, leading to up to eight different electronic states at low temperatures, hence, Gd-based systems have been proposed as qudits[33,42,49–51]. We employ isotopically enriched $^{160}$Gd$^{3+}$ ($I = 0$) to minimise the complexity of the study. The double-decker

complex crystallises in regular red block crystals in the tetragonal *P4/nmm* space group with half of the molecule per asymmetric unit, while two molecules reside in the unit cell (Supplementary Fig. 1). Locally, the Gd$^{3+}$ ion possesses approximate *D$_{4d}$* coordination.

A [Xe]4$f^7$ electronic configuration characterises the Gd$^{3+}$ ion embedded in Et$_4$N[$^{160}$GdPc$_2$], which in the Russell-Saunders coupling scheme leads to the $^8S_{7/2}$ ground state separated by more than $30 \times 10^3$ cm$^{-1}$ from the first exited $^6P$ multiplet. The ground $^8S_{7/2}$ state is isotropic, however, the degeneracy can be removed only by an odd perturbation, e.g., the coupling to an external magnetic field through the Zeeman term ($\mathscr{H}_Z = g\mu_B\mathbf{BS}$ where $g \approx 2$ is the value of the free electron). For the pure $^8S_{7/2}$ state, an isotropic $g$ value is expected. Nevertheless, when a small admixture with the first excited state $^6P$ is present, the ground state can be written as:

$$\sqrt{(1-\alpha^2)}\,{}^8S_{7/2} + \alpha\,{}^6P_{7/2} + \dots \qquad (1)$$

with the corresponding g-value being:

$$(1-\alpha^2)g({}^8S_{7/2}) + \alpha^2 g({}^6P_{7/2}) + \dots \qquad (2)$$

where $\alpha \sim 10^{-3}$. Spin-orbit admixture in the ground state is responsible for the ligand field splitting effects around two orders of magnitude smaller than $\langle L \rangle \neq 0$ in lanthanide SMMs. The Steven's operators formalism allows to model the ligand field interaction (3):

$$\mathscr{H}_{lf} = \sum_{n=1}^{3} B_{2n}^0 O_{2n}^0 + B_4^4 O_4^4 + B_6^4 O_6^4 \qquad (3)$$

where $O_q^k$ are Steven's operators and $B_q^k$ are the ligand field parameters. Note that the ligand field Hamiltonian contains only even terms in order to account for time-reversal symmetry. The axial terms, $B_{2n}^2 O_{2n}^2$, are invariant with respect to the point symmetry of the Gd$^{3+}$ site ($D_{4d}$), while the non-axial terms $B_4^4 O_4^4$ and $B_6^4 O_6^4$ are the result of the deviation from the ideal square antiprismatic symmetry of the molecule. Further symmetry-breaking requires an additional orthorhombic term $B_2^2 O_2^2$ in (3). The complete Hamiltonian for Et$_4$N[$^{160}$GdPc$_2$] is therefore the sum of the Zeeman and ligand field interaction:

$$\mathscr{H} = g\mu_B\mathbf{BS} + \sum_{n=1}^{3} B_{2n}^0 O_{2n}^0 + \left( B_2^2 O_2^2 + B_4^4 O_4^4 + B_6^4 O_6^4 \right) \qquad (4)$$

### μSQUID studies on Et$_4$N[GdPc$_2$]

To initiate our study, we first investigate the magnetic properties of the Et$_4$N[$^{160}$GdPc$_2$] complex via μSQUID magnetometry on micrometer-sized monocrystals of Et$_4$N[$^{160}$GdPc$_2$] diluted in the isostructural diamagnetic matrix of Et$_4$N[YPc$_2$] with [Gd/Y] ratio of 5%. The diluted complex crystallises in the tetragonal unit cell, *P4/nmm* space group, as confirmed by single crystal X-ray studies. The crystals were placed on an array of μSQUIDs and thermalised to sub-kelvin temperatures with the field applied along the easy axis of the molecules. Figure 1c,d show the temperature dependence of the hysteresis loops measured at a sweeping rate of 128 mT/s. The loops showcase the uniaxial character of the complex with the transition from open hysteresis loops to a superparamagnetic behaviour happening at the blocking temperature $T_b \sim 0.3$ K (See Supplementary Fig. 2). Above $T_b$, phonons possess enough energy to induce over-barrier relaxation of the molecular spin.

At zero magnetic field, a sharp transition is observed corresponding to quantum tunnelling of magnetisation (QTM). Note that, the Gd$^{3+}$ ion is characterised by a ground state with a half-integer spin $S = 7/2$, hence, according to Kramer's theorem, the ground state doublet $m = \pm 7/2$ should be degenerate. It is usually assumed that the coupling to environmental spins, both of electronic and nuclear origin,

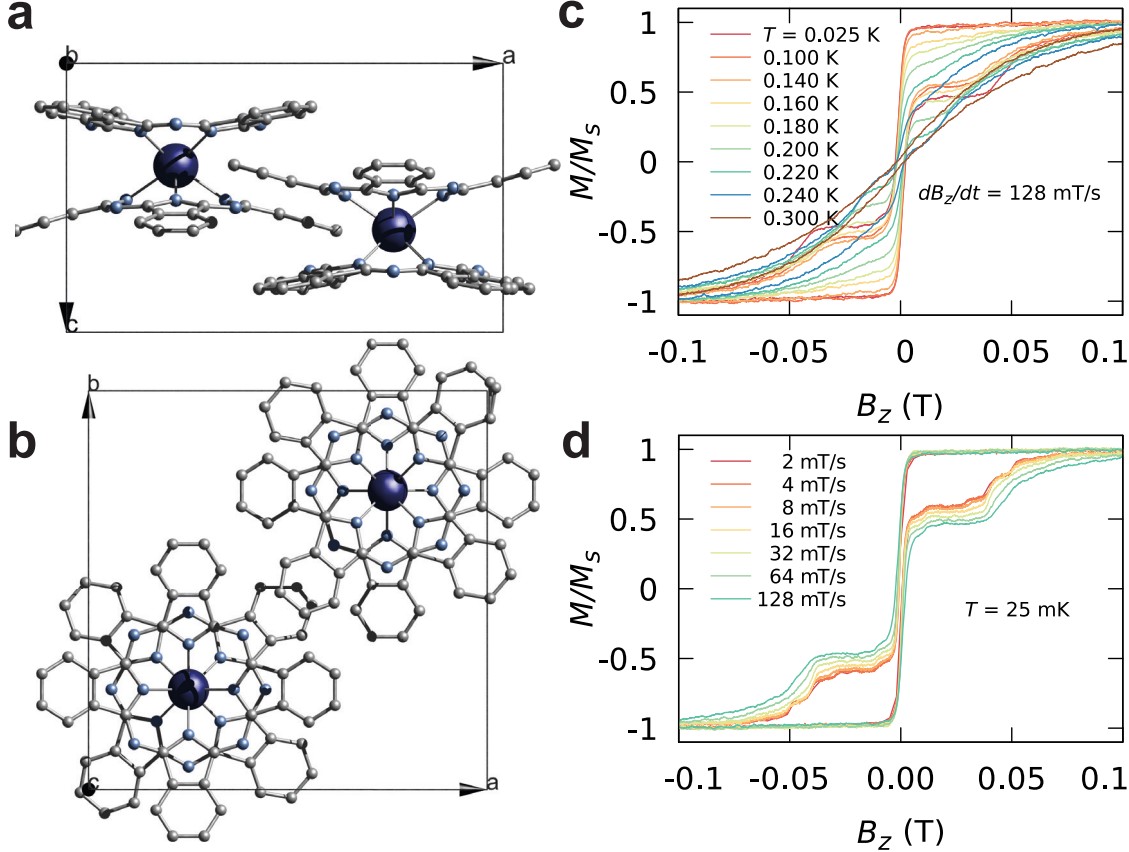

**Fig. 1 | Crystal structure of Et4N[160GdPc2] and μSQUID loops. a** Side view and **b** top view of the unit cell of $Et_4N[^{160}GdPc_2]$ showing two molecules residing within the unit cell. Hydrogen atoms have been removed for clarity. Colour code: Gd, dark blue; O, red; N, cyan; C, grey. **c** Hysteresis loops of the isotopically purified 5%

$Et_4N[^{160}GdPc_2]@Et_4N[YPc_2]$ determined by μSQUID measurements between 0.025 and 0.3 K and at a fixed sweeping rate of 128 mT/s and **d** at a fixed temperature of 25 mK and different sweeping rates.

breaks Kramer's degeneracy and hence makes the relaxation through QTM possible. Notably, in Fig. 1d the emergence of the fine structure in the magnetisation curves at small sweeping rates is observed. In order to investigate the origin of the fine structure, the temperature dependence of the magnetisation and their derivatives, in the temperature range where large variations occur [$T < 0.1$ K], were investigated at a sweeping rate of 8 mT/s (Fig. 2). A first distinction can be noticed between steps present at the lowest temperature, $T \approx 25$ mK (red-dotted lines), and the steps that appear with increasing $T$ (black-dotted lines). The temperature dependence of these steps suggests that the red-dotted line transitions occurring at 0, 0.048, and 0.067 T, depend only upon the ground state population, while transitions marked by the black-dotted lines are related to the population of the excited states, which becomes non-zero at higher temperatures.

Up to this point, the μSQUID data and the fine structure allows an initial determination of the spin Hamiltonian parameters given by Eq. 4. The blocking temperature to the superparamagnetic behaviour can be related to the zero-field splitting parameter through $k_B T_B \approx 3(|B_2^0|S^2 - |B_2^0|/2)$ [for half-integer spins], which allows the estimation of the axial zero-field splitting term leading to $|B_2^0| = 8.5$ mK. This value compares well with the expected value for ligand field splitting of the S-state ions. $|B_2^0| = 6.017$ K for $[TbPc_2]^-$ molecule, which is around three orders of magnitude larger than the value estimated for the $Et_4N[^{160}GdPc_2]$. The steps at 0.048 and 0.067 T can also be ascribed to crossings between the ground state $|+7/2\rangle$ and first excited states $|-5/2\rangle$ and $|-3/2\rangle$, respectively. The transitions that are not observed at $T = 25$ mK can be ascribed to collective dynamics[52], and not

due to the direct anticrossing with the excited levels due to the lack of equivalent transitions at negative fields. Note that, the relaxation in zero field does not induce excitations from $|\pm 7/2\rangle$ to $|\pm 5/2\rangle$ (hence explaining the new steps) because ground state quantum tunnelling dynamics is a non-dissipative process.

A more accurate determination of the anisotropy parameters from μSQUID loops can be achieved by considering that the anisotropy is dominated by the axial term, as it usually is the case for uniaxial SMMs. Equation (5) relates the field value of the level crossings between $|m\rangle$ and $|m'\rangle$ and the Steven's coefficients $B_2^0$ and $B_4^0$:

$$H_n = \frac{3nB_2^0}{g\mu_B}\left[1 + \frac{B_4^0}{3B_2^0}((m-n)^2 + m^2)\right] \tag{5}$$

in (5), $n = m + m'$ is the order of the level crossing. By adjusting the off-diagonal terms $B_2^0 O_2^0$ and $B_4^0 O_4^0$ the thermally excited steps can be adjusted to correspond to spin-spin cross-relaxation processes. Figure 2c shows the corresponding Zeeman diagram adjusted to the fines structure of $Et_4N[^{160}GdPc_2]$. It is important to stress that the obtained set parameters are not unique and depend on the assumptions made on the origin of the steps. In fact, all the steps can originate from collective processes (implied by the relative magnitude of the steps at zero/non-zero field), which would lead to different sets of parameters. Thus, additional information is required to unambiguously determine the ligand field coefficients, as μSQUID on its own, for complex systems, is not sensitive enough to obtain quantitative information on the off-diagonal terms.

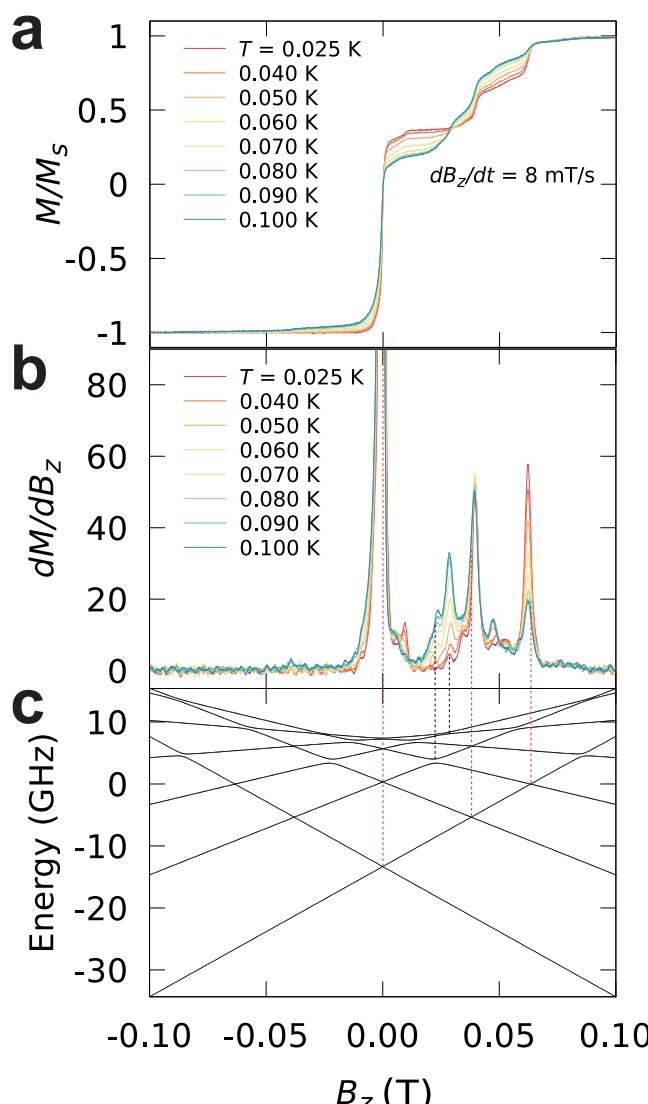

**Fig. 2 | μSQUID loops. a** Temperature dependence of the magnetisation curves and **b** their derivatives for $T < 0.1$ K, at a fixed sweeping rate of 8 mT/s. The red-dotted lines indicate the relaxation steps present even at the lowest temperature of $T \approx 25$ mK, while the black-dotted lines mark the steps that appear as $T$ increases. **c** The Zeeman diagram obtained by diagonalisation of the Spin Hamiltonian given by Eq. 4 with $g = 2$, $B_2^0 = -5.2 \times 10^{-1}$ GHz, $B_2^2 = 1.0 \times 10^{-1}$ GHz and $B_4^4 = -3.5 \times 10^{-4}$ GHz.

### μSQUID-EPR studies on Et₄N[GdPc₂]

In order to unambiguously and quantitively study the ligand field, μ-SQUID was employed as a magnetic probe in an EPR experiment. The technique consists in excitation of the spin system with microwave pulses during the collection of the μ-SQUID loops. This technique has been employed to study resonant photon absorption in $S = \frac{1}{2}$ systems, $Cr_7Ni$ and $V_{15}$ and to investigate photon-assisted tunnelling in $Fe_8$[53–55], however, herein the technique has been optimised for the determination of high-order ligand field parameters. Electromagnetic pulses were applied to the crystal in the μ-SQUID array by employing a frequency synthesiser while maintaining the temperature at 40 mK. Figure 3a shows magnetisation curves as a function of frequency obtained while scanning the magnetic field with a fixed sweeping rate of 8 mT/s and by applying radiofrequency pulses of 40 μs width and a period of 300 μs in between μSQUID measurements (See Supplementary Fig. 4). Absorption dips in the magnetisation curves, denoting resonant transitions following the resonant condition: $h\nu = |E(m) - E(m')|$, can be observed. In Fig. 3b,c,

the data are more clearly visualised by applying a Gaussian filter to extract the absorption peaks and by plotting the set of measurements for $B \in [-0.5 : 0.1]$ T and $\nu \in [1 : 35]$ GHz as a colour map where the intensity of the colour stands for the magnitude of the peak.

The SMM behaviour with pseudo uniaxial magnetic anisotropy was confirmed by the resonance maps. For better understanding, we label the different transitions by $(m, m')$ where $m$ and $m'$ designate the eigenvalues of $S_z$ operator: $S_z|m\rangle = m|m\rangle$. This notation works well for the transitions in high longitudinal magnetic fields and for large $|m + m'|$ values where the axial approximation holds.

The transition lines can be grouped in sets characterised by similar absolute values of the slope. Each group corresponds to transitions with a different selection rule. Most of the transition lines are part of the set with the smallest absolute value of the slope (denoted by **1, 2, 3,..., 6** in Fig. 3b) and correspond to the "allowed' dipolar transitions characterised by the selection rule $\Delta m = \pm 1$. The magnitude of the zero-field splitting obtained from μSQUID measurements, the linearity and the relative intensity of different lines, indicate that **1** corresponds to the (7/2, 5/2) transition. The subsequent lines result from (5/2, 3/2), (3/2, ½), (1/2, −1/2), (−3/2, −1/2) and (−5/2, −3/2) transitions. Moreover, **7, 8**, and **9** correspond to higher-order transitions allowed by the non-axial interactions. Additional features in the resonance map indicate the presence of interactions that break the axial symmetry and deviate from linearity at small fields, $|B| < 0.2$ T, and the direct observation of level anticrossings, marked with coloured circles in Fig. 3c.

Employing the uniaxial nature of the system, the magnitude of the g-value can be evaluated by fitting the linear (high field region) part of the spectrum resulting in:

$$g = \frac{h}{\mu_B} \frac{d\nu}{dB} = 1.96(1) \tag{6}$$

As observed, the obtained experimental value ($g = 1.9(1)$) differs significantly from the theoretical prediction (2.0023), indicating a strong contribution of the excited state.

### Control over the non-axial interactions

One of the most important advantages of the μSQUID-EPR technique is the control over the non-axial spin Hamiltonian through the simultaneous application of transverse magnetic fields. This allows to (i) maintain the direction of the applied field fixed and vary the frequency at different constant applied transverse fields; and (ii) fix the frequency and vary the direction of the applied magnetic field.

Figure 4a shows the resonance map where the field was swept along the easy axis with an additional applied constant transverse field of 20 mT. The effect of the transverse field is directly observed as an increase in the magnitude of the tunnel splittings. Figure 4b shows the resonant map while sweeping along different directions in the (easy-hard) plane at a constant frequency of 17.6 GHz. Importantly, the angular dependence allows the determination of the signs of the off-diagonal terms, whereas the tunnel splittings give access only to the magnitude of the ligand field coefficients.

### Discussion

#### Axial ligand field interaction

In order to interpret the observed behaviour, we begin our analysis with an initial evaluation of the diagonal ligand field terms by assuming an axial approximation. Extrapolating the linear high field region of the (7/2, 5/2), (5/2, 3/2), (3/2, 1/2) transitions to zero field allows the determination of the zero-field resonant (ZFR) splittings, denoted by $\nu_{01}$, $\nu_{02}$, and $\nu_{03}$. Employing the algebraic form of three axial $O_{2n}^0$ Steven operators, the following relations between ZFR values and the Steven's

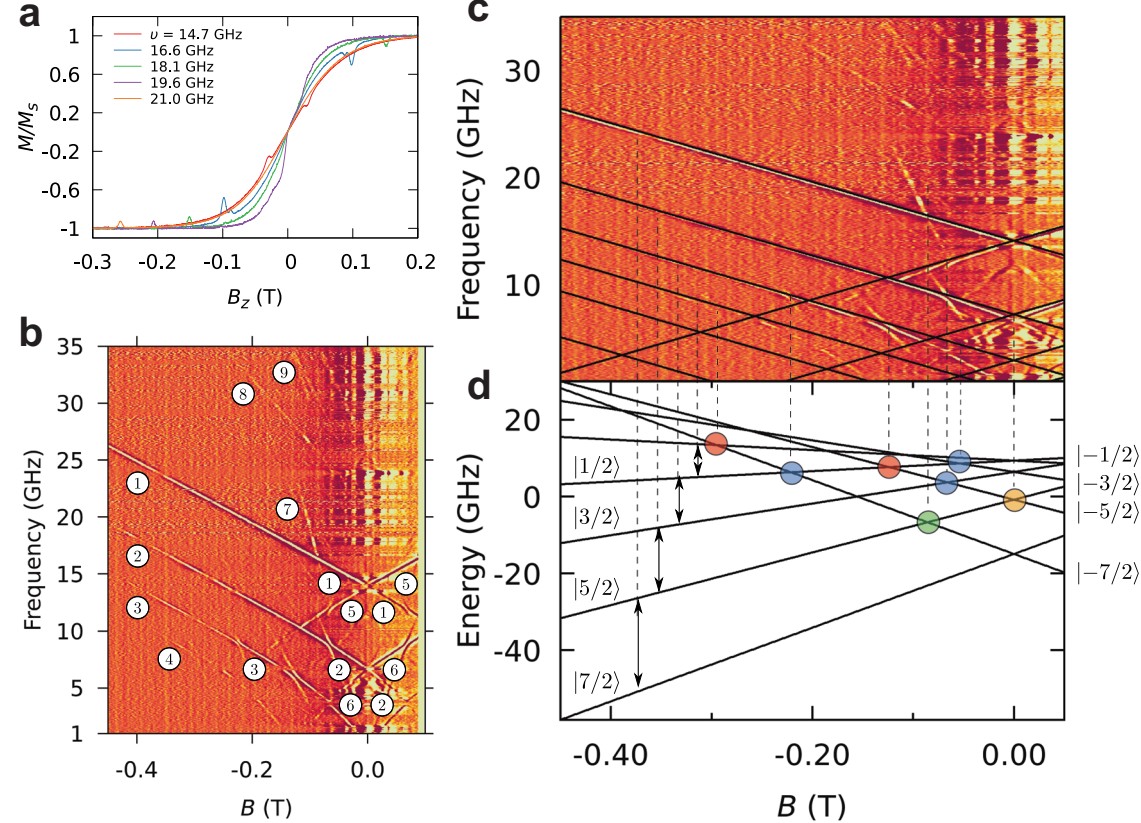

**Fig. 3 | μSQUID loops EPR. a** Magnetisation curves as a function of the frequency of the RF source obtained while sweeping the magnetic field with a fixed sweeping rate of 8 mT/s and by applying pulses of 40 μs width and 300 μs period. **b** Resonant maps obtained by sweeping along the easy axis with a constant sweeping rate of 8 mT/s and applying RF pulses with 40-μs width every 300 μs. The labels correspond to $(m, m')$ transitions (see the main text). **c** Linear fit of the resonance map by

using: $g = 1.96$, $B_2^0 = -6.83 \times 10^{-1}$ GHz, $B_4^0 = -1.5 \times 10^{-3}$ GHz, $B_6^0 = 1.6 \times 10^{-8}$ GHz obtained from Eqs. 7–9. $(m, m \pm 1)$ dipolar transitions are shown as black lines superimposed on the resonance map. **d** Zeeman diagram with marked level anticrossing, according to the selection rules: $\Delta m = 4$ (blue), $\Delta m = 6$ (green), $\Delta m = 5$ (yellow), and $\Delta m = 3$ (red).

coefficients are obtained, allowing to solve exactly for $B_{2n}^0$:

$$h\nu_{01} = |-6B_2^0 + 720B_4^0 - 17640B_6^0| \qquad (7)$$

$$h\nu_{02} = |-12B_2^0 + 600B_4^0 + 17640B_6^0| \qquad (8)$$

$$h\nu_{03} = |-18B_2^0 + 1200B_4^0 - 7560B_6^0| \qquad (9)$$

The eigenvalues of the axial Hamiltonian as a function of the applied magnetic field are shown in Fig. 3c and the respective $(m, m \pm 1)$ dipolar transitions are shown as black lines superimposed on the resonance map. Good agreement with the experimental data is obtained, indicating that the hypothetical axial symmetry is a good approximation for Et₄N[¹⁶⁰GdPc₂], and confirming the indexation of different transition lines. Furthermore, the higher-order transitions in Fig. 3b can be identified. For example, **7** corresponds to the transition (+7/2, −7/2) and **8, 9**, are (7/2, 3/2) and (7/2, 1/2), respectively.

**Transverse field interaction**
When considering the off-diagonal interactions in (4) the eigenvectors of $\mathscr{H}$ are no longer eigenvector of $S_z$ but instead they should be written as a linear combination of the $|m\rangle$ states. The exact analytical solution to the eigenvalue-eigenvector problem of a generalised spin Hamiltonian is often difficult to find and a perturbative approach is preferred.

Two states, $|m\rangle$ and $|m'\rangle$, mixed by the transverse term of order $k$, $BS_\pm^k$, result in the following expression for the tunnel

splitting:

$$\triangle_m^{m'} \sim B_2^0 S^2 \left[ BS^k / \left( 2B_2^0 S^2 \right) \right]^{(m'-m)/k} \qquad (10)$$

Note that the spin parity effect, viz., the mixing between levels $|m\rangle$ and $|m'\rangle$, is possible only if the difference $|m-m'|$ is a multiple $k$. Consequently, the admixing of different states is mostly significant for levels at the top of the barrier (small absolute value of $m$) and at level crossings where the axial and transverse interactions are comparable in magnitude. This makes the repelling regions in the parallel Zeeman pattern a central feature for exploring non-axial interactions.

In Fig. 3c, the anticrossings are highlighted with circles in the Zeeman diagram. Two repelling regions in the transition map correspond to the same anticrossing in the Zeeman diagram. The selection rule involved in the mixing of the energy levels corresponds to $\Delta m = 4$ (blue), $\Delta m = 6$ (green), $\Delta m = 5$ (yellow), and $\Delta m = 3$ (red). These typically forbidden EPR transitions provide the base for the determination of high-order parameters[42].

**Experimental complete spin hamiltonian parameters**
With the complete set of data, we are now in a position to evaluate the spin Hamiltonian of Et₄N[¹⁶⁰GdPc₂]. Due to the increased transverse field, a significant level mixing of the states occurs and the $(m, m')$ indexation does not characterise the transitions. Instead, the observed transitions are described by the order of the eigenvalue and the corresponding selection rule: $(k, \Delta m)$ in Fig. 4c, d. The initial set of

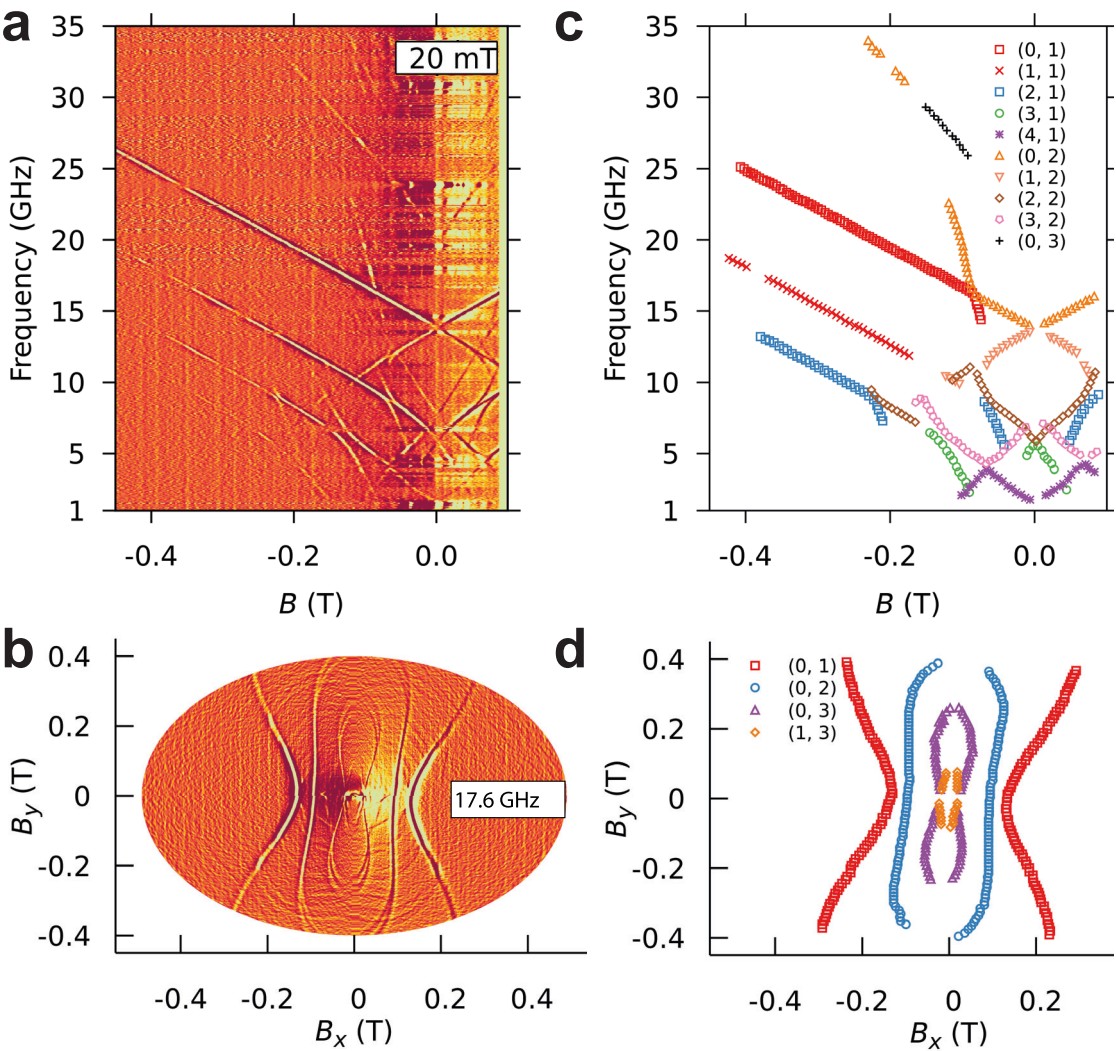

**Fig. 4 | μSQUID-EPR. a** The resonance map when the field was swept along the easy axis with an additional, constant transverse field of 20 mT. **b** The resonant map while sweeping different directions in the (easy-hard) plane at a constant frequency of 17.6 GHz. **c, d** show the sampling of the resonance maps. $(k, \Delta m)$ denotes an individual transition with $k$ being the order of the eigenvalue and $\Delta m$ the corresponding selection rule.

parameters, $B_k^0$, is obtained from the analysis made in the axial approximation, while the transverse ligand field coefficients are taken to be of the same order as diagonal terms. Subsequently, diagonalisation of the spin Hamiltonian is carried out at the experimental points with $g$, $\varphi$, $B_2^0$, $B_4^0$, $B_6^0$, $B_2^2$, $B_4^4$ as fitting parameters, where $\varphi$ indicates a misalignment angle between the applied field and the easy axis of the molecule. Note that the tunnel splitting, $\Delta$, is the result of the linear combination of $O_4^4$ and $O_6^4$ interactions, without direct means to distinguish between them; hence, in the following discussion $B_4^4O_4^4$ is the only term that will be considered as it incorporates the $B_6^4O_6^4$ component. All the other observed tunnelling gaps cannot be explained by the direct application of $B_4^4O_4^4$ and a combination of odd and even transverse interactions has to be employed. For example, mixing of $|+5/2\rangle$ and $|-5/2\rangle$ states with $\Delta m = 5$ is possible only if one includes an environmental magnetic field, while the anticrossings with $\Delta m = 3$ or 6 indicate the orthorhombic interaction $B_2^0O_2^0$.

Figure 5 and S3 show the simultaneous fit (black lines) of both the transverse field maps and angular maps, with $g = 1.96$, $\varphi = 2.8\circ$, $B_2^0 = -6.80 \times 10^{-1}$ GHz, $B_4^0 = -1.57 \times 10^{-3}$ GHz, $B_6^0 = 1.6 \times 10^{-7}$ GHz, $B_2^2 = -2.75 \times 10^{-1}$ GHz and $B_4^4 = 3.38 \times 10^{-3}$ GHz. The resonance lines broaden and then split at large transverse fields due to the presence of two inequivalent Gd$^{3+}$ centres with a slight angle between their easy axes. This is consistent with the two Et$_4$N[$^{160}$GdPc$_2$] molecules in the

unit cell observed through crystallographic analysis. Furthermore, a misalignment angle of 2.8° explains the mixing of levels with an odd selection rule: $\Delta m = 3$ and $\Delta m = 5$.

After having determined the high-order transverse ligand field, we are in a unique position to compare the experimental parameters with those obtained via theoretical means. As observed in Supplementary Tables 1–3, we find that although the magnitude of the parameters is correct, the numerical values are different, indicating that when high precision in the parameters is desired, CASSCF calculations fail.

Herein we provide access to an experimental technique that allows the direct determination of high-order ligand field parameters, by simultaneously measuring the magnetisation while performing multifrequency EPR studies. The application of transverse magnetic fields, while performing the measurements, allows us to modulate the non-axial spin Hamiltonian and thus determine the symmetry-breaking interactions that are present in the system. The high resolution of the resonance maps in 4D ($B_x$, $B_y$, $B_z$, ν) allows for a precise determination of the ligand field parameters. A comparison of the experimentally determined ligand field parameters with the theoretically calculated values shows the need for experimental validation in future attempts to improve the theoretical models. Further development of the EPR-μSQUID technique promises to provide deep insights into the central

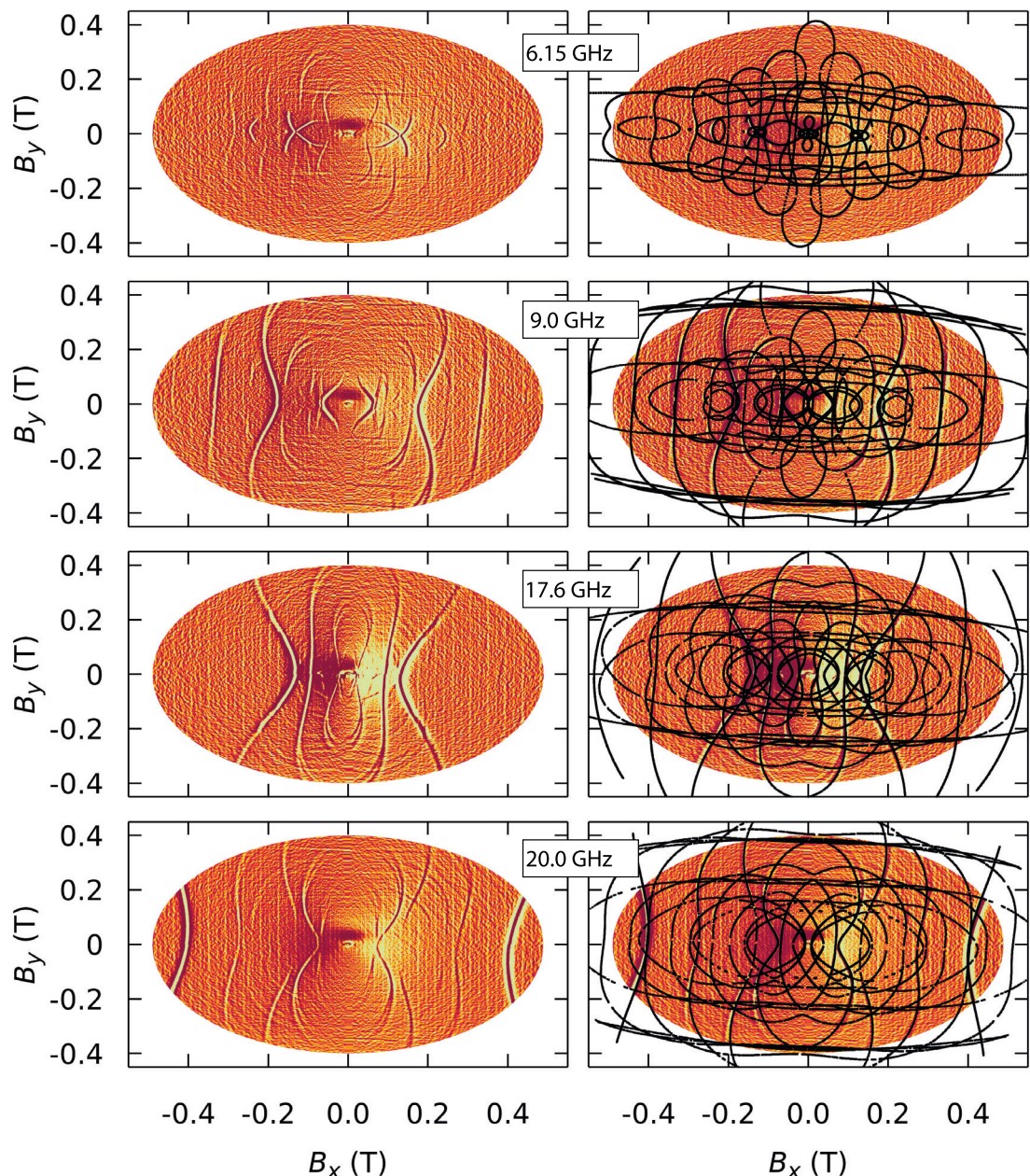

**Fig. 5 | µSQUID-EPR.** (left) Resonance maps obtained by sweeping the angle at the fixed frequencies of 6.15 GHz, 9 GHz, 17.6 GHz, and 20 GHz. (right) Fit of resonance maps giving $g = 1.96$, $\varphi = 3°$, $B_2^0 = -6.80 \times 10^{-1}$ GHz, $B_4^0 = -1.57 \times 10^{-3}$ GHz, $B_6^0 = 1.6 \times 10^{-7}$ GHz, $B_2^2 = -2.75 \times 10^{-1}$ GHz and $B_2^4 = 3.38 \times 10^{-3}$ GHz.

actors that determine the anisotropy of SMMs and subsequently facilitate chemical tunning of molecular systems for quantum information processing applications.

## Methods

### µSQUID apparatus

Low-temperature magnetisation measurements in the range of 0.30–5.0 K were performed on single crystals of Et$_4$N[GdPc$_2$] using a µSQUID apparatus at different sweep rates between 0.280 and 0.002 T s$^{-1}$. The time resolution is -1 ms. The magnetic field can be applied in any direction of the µSQUID plane, employing a 3D vector magnet, with precision better than 0.1° by separately driving three orthogonal coils. In order to ensure good thermalisation, each sample was fixed with apiezon grease.

### EPR-µSQUID setup

The EPR-µSQUID loops of the system were obtained by resonant excitation of the system *via* RF pulses (frequency, power, width, etc.,) applied while performing the µSQUID measurements. As the µSQUID cannot function under RF irradiation, the RF pulses applied to the sample were interleaved with the current pulses used to trigger the µSQUID (see Supplementary Fig. 4). To generate electromagnetic radiation, a AnritsuMG369x frequency synthesiser triggered by a pulse generator was employed to generate EM pulses with a width ranging from nanosecond to continuous radiation with powers up to 20 dBm. The generated signal is transmitted through a coaxial microwave cable which is thermalized between room temperature and 40 mK stage of the cryostat and then applied to the sample through a wire suspended above the crystal. The RF magnetic field is roughly perpendicular to

the easy axis of the crystal. The power is adjusted for each frequency to maximise the absorption of the RF signal.

### g-value

The main contributions to g-value for S-state ions are: (i) quantum electrodynamical value for a free electron (2.0023); (ii) mixing of $^6P$ exited state in the ground state $^8S_{7/2}$ (−0.0078); and (iii) Judd and Lindgren relativistic contributions (−0.0017), leading to a final theoretical value of $g = 1.992(8)$.

### Fitting

The square deviation between the theoretical prediction and experimental points was evaluated by: $\chi^2 = \sum (\nu_{exp} - \nu_{SHP})^2$ and $\chi^2$ was minimised in an iterative process by using the Marquardt−Levenberg nonlinear algorithm with: $g, \varphi, B_2^0, B_4^0, B_6^0, B_2^2, B_4^4$ as fitting parameters. See SI section D for details.

### Data availability

Supplementary information is available in the online version of the paper. Full crystallographic details for the complexes can be found in CIF format: see the Cambridge Crystallographic Data Centre database (CCDC 2192851-1547623). The experimental data for Figs. 4a, c and 5 generated in this study have been deposited in the Figshare database under accession code https://figshare.com/s/af2dc483e82240f8d835. Source data are provided with this paper.

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

## Acknowledgements
We acknowledge the DFG-CRC 1573 "4 f for Future" and the Karlsruhe Nano Micro Facility (KNMF, www.kit.edu/knmf) for the provision of access to instruments at their laboratories. EMP thanks the Panamanian National System of Investigators (SNI) and SENACYT (project PFID-FID-2021-60) for support. WW thanks the A. v. Humboldt Foundation and the ERC grant MoQuOS No. 741276.

## Author contributions
W.W. conceived the idea and supervised the project. G.T., W.W., and M.S. created the EPR-µSQUID setup. G.T. and E.B. collected, processed, and simulated the data and interpreted the results with W.W., M.R., and E.M.P. designed the complex and synthesised and characterised the samples. The manuscript was written by G.T., E.M.P., and M.S. with input from all authors.

## Funding

## Competing interests
The authors declare no competing interests.
