## [Peer Review File · Nature Communications]

Reviewers' Comments:

Reviewer #1:

Remarks to the Author:

Review of "Direct Determination of High Order Transverse Ligand Field Parameters *via* μ SQUID-EPR in a Et₄N[160GdPc₂] SMM"

G. Taran et al., have reported determination of ligand field parameters using microSQUID-EPR experiment. I believe the experimental technique of combining microSQUID with EPR and operating the experiment at ultralow temperature is highly challenging and two-dimensional map of spin resonances are very nicely measured. However, I do not think that this work meets the high standard of novelty required by Nature Communications in following reasons.

First of all, in the introduction of the manuscript, the authors have emphasized about quantum computing, SMM based quantum technologies and even quantum algorithms. However, I found out that main text and conclusion section of the manuscript does not contain any quantum technology and information related results and data. Basically, the manuscript just shows a measurement of magnetic anisotropic parameters using micro-SQUID-EPR and Hamiltonian fitting routine without any uses of this particular SMM toward quantum information. Moreover, there have been many studies of SMM molecules using EPR, SQUID, and STM (even having a spatial resolution of individual atoms and molecules), which can determine magnetic anisotropic parameters very precisely in 3-dimension (ex, S. Yan et al., *Nano Lett* **15**, 1938 (2015)).

I believe that the authors may emphasize finding higher order transverse terms using this technique. However, what will be interesting, important, and unique perspective of this particular SMM and/or what will be important aspect of finding higher order transverse terms? I am uncertain whether the authors have addressed those in the manuscript.

Secondly, I found out that several discussions in the manuscript is unclear to me and I think the authors need to explain more.

- There are 72 references in introduction part and I am uncertain that all of those references are related to this manuscript. On the other hand, there are only 5 references for the main text. Please comment on this.

- In 2nd subtopic of the result section (pg 3), GdPc₂ complexes for this study are the micrometer sized monocrystals. Have the authors confirmed that the sample has well-ordered unit cell as shown in Fig. 1a,b via crystal analysis tool (X-ray, etc)?

- In the same paragraph (pg 3-4), the authors mentioned that "The loops showcase the uniaxial character of the complex, while transition from open hysteresis loops to a superparamagnetic behaviour occurs at the blocking temperature $T_b \sim 0.3K$. Above T_b phonons possess enough energy to induce over-barrier relaxation of the molecular spin."

→ When I closely look into the hysteresis loop in Fig. 1c, I see that the loop is still open at 0.3K. If the material becomes paramagnetic, the loop has to be closed. Please comment on this. Also, I think the authors should show the data at higher temperatures (>0.3K) as well.

- In the next paragraph (pg 4), the authors mentioned that “A first distinction can be noticed between steps present at the lowest temperature, $T \approx 25$ mK (red dotted lines), and the steps that appear with increasing T (black dotted lines). The temperature dependence of these steps suggests that the transitions at 0.048 and 0.067 T are contingent only upon the ground state population, with the remaining transitions being related to the population of the excited states at higher temperatures.”

→ For 0.048 and 0.067T, which peaks in Fig. 2b do the authors are mentioning? And It is difficult for me to understand what the authors means like “contingent” and what are the remaining transitions in Fig. 2b?

- In the next paragraph (pg 4-5), the author mentioned that “Up to this point, the μ SQUID data and the fine structure allows an initial determination of the spin Hamiltonian parameters given by Eq. 4. $k_B T \approx 3(|B_0(2)|S_2 - |B_0(2)|/2)$ allows the estimation of the axial zero-field splitting term leading to $|B_0(2)| = 8.5$ mK”

→ How can the data of Fig. 1, 2 and eq. 4 determine the parameters? And how does the value of 8.5mK come from?

- In the first paragraph of pg 7, the authors mentioned “Figure 3a shows magnetisation curves as a function of frequency obtained while scanning the magnetic field with a fixed sweeping rate of 8 mT/s and by applying radiofrequency pulses of 40 μ s width and a period of 300 μ s in between μ SQUID measurements.”

→ The authors used microwave pulse with 40us and 300us period. Why do the authors choose those time scales? Have you changed to different time scale? Because, for the spin resonance experiment, when you saturate the transition, the absorption signal can mislead for determination of the transition probabilities. Also, I see that you sweep RF in very wide range of high frequencies (upto 21GHz). And in this case, I believe that your microwave line could have very different rate of loss across such a wide range. This means that the RF amplitude over freq sweep can be highly frequency-dependent. How do the authors calibrate this?

- In the second paragraph of pg 8, the authors mentioned “Most of the transition lines are part of the set with the smallest absolute value of the slope (denoted by **1, 2, 3**,... in Fig. 3b) and correspond to the “allowed” dipolar transitions characterised by the selection rule $\Delta m = \pm 1$. The magnitude of the zero-field splitting obtained from μ SQUID measurements, the linearity and the relative intensity of different lines, indicate that **1** corresponds to the (7/2, 5/2) transition. The subsequent lines result from (5/2, 3/2), (3/2, 1/2), ..., transitions. Moreover, **7, 8, and 9** correspond to higher-order transitions allowed by the nonaxial interactions.”

→ what are the transitions for the denoted numbers of **4, 5, 6**? If I follow the authors’ notation, **4** = (1/2, -1/2), **5** = (-1/2, -3/2), **6** = (-3/2, -5/2). Is this correct? Please explain the transition for **4, 5, 6**.

Lastly, I found out many basic typos and grammatic mistakes.

- In line 7, 2nd paragraph of pg 2:

The possibility of chemically modify the periphery ... → The possibility of chemically modifying the periphery ...

- In line 2, after the equation (2) of pg 3:

Spin-orbit admixture in the ground state is responsible ... smaller than in $\langle L \rangle$... → Spin-orbit admixture in the ground state is responsible ... smaller than one in $\langle L \rangle$...

- In line 4, 1st paragraph of pg 5:

0.048 and 0.067mT → 0.048 and 0.067 T

- In line 14, 1st paragraph of pg 7:

for magnitude of the peak Fig. 3. → for magnitude of the peak in Fig. 3.

- In line 6, caption of the Fig. 3 of pg 7:

... Obtained from Eq. 5.7. $(m, m \pm 1)$... → Eq 5, 7

- In line 1, 1st paragraph of pg 10:

Allowing to solve exactly for $B_0(2n)$. The eigenvalues of the axial Hamiltonian as a function of the applied magnetic field are shown in Fig. 3c and ...

→ Allowing to solve exactly for $B_0(2n)$, the eigenvalues ...

- In line 1, 5th paragraph of pg 10:

In Fig. 3c the anticrossings are highlighted...

→ In Fig. 3c, the anticrossings are highlighted...

- In line 4, 5th paragraph of pg 10:

corresponds to $\Delta m = 4c$ (blue), ...

→ corresponds to $\Delta m = 4$ (blue), ...

- In line 5, 6th paragraph of pg 10:

selection rule: $(k, \Delta m)$ Fig. 4c,d. ...

→ selection rule: $(k, \Delta m)$ in Fig. 4c,d. ...

- In line 12, 6th paragraph of pg 10:

hence, in the following discussion $B_4(4)O_4(4)$ is the only term that will be considered it incorporates the $B_4(6)O_4(6)$ component.

→ I cannot understand this sentence grammatically.

Due to those reasons listed above, I cannot recommend this manuscript to be published in Nature Communication.

Reviewer #2:

Remarks to the Author:

The manuscript presents the use of mu-SQUID-EPR for the estimation of the transverse crystal field parameter in SMM. The knowledge of these parameter is very useful in the domain of quantum information processing since a weak variation of bad knowledge of it dramatically change the dynamics property of the system.

The results presented in the manuscript is of high interest. The manuscript is clear, well written and made for a broad audience.

The author manages to clearly explain very specific concept in spectroscopy which made the manuscript suitable for Nature audience.

I have no major concern on it but I have a few suggestions

1- The manuscript is made for broad audience but on the other hand I think there is a lack of detail for specialist in the field. Although not necessary I think the numeric data of the figure 4 ac and 5 should be provided to allow the reproducibility of the fit process

2- The fitting and simulation procedure explanations is a bit weak. I suggest the author to add detail (may be in the SI)

3- The valus of Bkl are given without error estimation. However I think the error estimation is very important since some Bkl parameters (like B44) are much more sensitive variation of the frequency of resonance than others

4- Mu-SQUID EPR is a very specific technic and the author add a recent achievement on Gd3+. <https://doi.org/10.1103/PhysRevApplied.18.014054>

In conclusion, the manbuscript should be published in Nature communication with the "optional" comments above

Reviewer #3:

None

Response to the Referees

Thank you very much for considering our manuscript (NCOMMS-22-33063-T) for publication. On behalf of all co-authors, I would like to thank the referees for taking the time to carefully review our manuscript and for their valuable comments and suggestions. We have followed all the comments and made revisions in the manuscript accordingly, which are highlighted in cyan. Our point-to-point responses to all questions and remarks are provided below. We hope that the referees will be satisfied with our revision and will agree that the manuscript is now acceptable for publication in *Nature Communications*. Thank you once more for your consideration.

Reviewer 1

G. Taran et al., have reported determination of ligand field parameters using microSQUID-EPR experiment. I believe the experimental technique of combining microSQUID with EPR and operating the experiment at ultralow temperature is highly challenging and two-dimensional map of spin resonances are very nicely measured. However, I do not think that this work meets the high standard of novelty required by *Nature Communications* in following reasons.

1. First of all, in the introduction of the manuscript, the authors have emphasized about quantum computing, SMM based quantum technologies and even quantum algorithms. However, I found out that main text and conclusion section of the manuscript does not contain any quantum technology and information related results and data. Basically, the manuscript just shows a measurement of magnetic anisotropic parameters using micro-SQUID-EPR and Hamiltonian fitting routine without any uses of this particular SMM toward quantum information. Moreover, there have been many studies of SMM molecules using EPR, SQUID, and STM (even having a spatial resolution of individual atoms and molecules), which can determine magnetic anisotropic parameters very precisely in 3-dimension (ex., S. Yan et al., *Nano Lett* **15**, 1938 (2015)).

I believe that the authors may emphasize finding higher order transverse terms using this technique. However, what will be interesting, important, and unique perspective of this particular SMM and/or what will be important aspect of finding higher order transverse terms? I am uncertain whether the authors have addressed those in the manuscript.

Answer: We thank the referee for the valuable comment. The purpose of the introduction was to show the reader the following aspects: (i) the importance of SMMs and their applicability in diverse technological applications, and (ii) that despite having several important techniques for the investigation of the anisotropy in SMMs, the precise determination of high-order parameters is still a challenge. Indeed, the referee is correct, and techniques such as EPR and SQUID have been employed to experimentally determine crystal field parameters. However, they provide limited sensitivity compared to the technique we report. Moreover, the μ SQUID-EPR technique provides the possibility to perform the measurement directly on a crystal assemble without requiring complex preparation procedures, as required for the STM-EPR technique, which might compromise the integrity of molecular systems, such as SMMs. We, hence, consider the μ SQUID-EPR technique to be a valuable easy-access technique for the determination of high order crystal field parameters at mK temperatures in bulk. Taking into consideration the valuable comment of the referee, we have modified the introduction to highlight the importance of SMMs as well as the determination of the ligand field parameters for their implementation in novel quantum technologies. The modifications to the introduction are all highlighted in cyan in the revised version of the MS.

2. Secondly, I found out that several discussions in the manuscript is unclear to me and I think the authors need to explain more. There are 72 references in introduction part and I am uncertain that all of those references are related to this manuscript. On the other hand, there are only 5 references for the main text. Please comment on this.

Answer: Again, we thank the referee for the comment. In the original form, the introduction tried to highlight the importance of SMMs in several technological applications, as well as to show the importance of the determination of the anisotropic parameter for the understanding, thus, their exploitation. In this regard, there are several important works towards the implementation of such systems in technological applications, which

we deemed important to quote. Considering the reviewer concern, the introduction has been shortened and modified as shown in the revised version of the MS. The references have also been modified and a shorter reference list is provided. Concerning the references in the main text, this is related to the novelty of the technique. Although we have reported the application of microwave pulses during the μ SQUID loops acquisition, this is the first report of detailed multifrequency-variable applied transverse fields study.

3. In 2nd subtopic of the result section (pg 3), GdPc₂ complexes for this study are the micrometer sized monocrystals. Have the authors confirmed that the sample has well-ordered unit cell as shown in Fig. 1a,b via crystal analysis tool (X-ray, etc)?

Answer: Indeed, during the synthetic standardisation procedure, X-ray crystallography was used as a critical technique to confirm the phase of the crystals. Both, Et₄N[¹⁶⁰GdPc₂] and Et₄N[YPc₂] samples were fully characterised by X-ray crystallography denoting the same packing. The diluted samples were also screened via X-ray crystallography to confirm the phase purity of the crystals, revealing that the sample crystallised in a single phase. Furthermore, our μ SQUID studies show the presence of a singly-oriented anisotropic specie, confirming the X-ray results. To support our statements, we have included the crystallographic data for the Et₄N[YPc₂] complex and a brief description of the MS.

4. In the same paragraph (pg 3-4), the authors mentioned that “The loops showcase the uniaxial character of the complex, while transition from open hysteresis loops to a superparamagnetic behaviour occurs at the blocking temperature $T_b \sim 0.3$ K. Above T_b phonons possess enough energy to induce over-barrier relaxation of the molecular spin.” When I closely look into the hysteresis loop in Fig. 1c, I see that the loop is still open at 0.3K. If the material becomes paramagnetic, the loop has to be closed. Please comment on this. Also, I think the authors should show the data at higher temperatures (>0.3 K) as well.

Answer: We thank the referee for the valuable comment. The purpose of Fig. 1c was to show the temperature dependence of the fine structure of the magnetisation curves, not the paramagnetic crossover. As highlighted by the referee, the loop at 0.3 K in Fig. 1c is still open. The reason for this is that the temperature at which the system becomes paramagnetic depends on the timescale of the experiment (field-sweeping rate). To observe the paramagnetic crossover of the system, hysteresis loops at lower sweeping rates are best suited. This is better shown in Fig. S2. As observed, at 32 mT/s sweeping rate, the system becomes paramagnetic above 0.3 K. This Figure has been added to the SI and referred to in the MS.

Figure S2: Temperature dependence of the magnetization curves at a fixed sweeping rate of 32 mT/s

5. In the next paragraph (pg 4), the authors mentioned that “A first distinction can be noticed between steps present at the lowest temperature, $T \approx 25$ mK (red dotted lines), and the steps that appear with increasing T (black dotted lines). The temperature dependence of these steps suggests that the transitions at 0.048 and 0.067 T are contingent only upon the ground state population, with the remaining transitions being related to the population of the excited states at higher temperatures.” For 0.048 and 0.067T, which peaks in Fig. 2b do the authors are mentioning? And It is difficult for me to understand what the authors means like “contingent” and what are the remaining transitions in Fig. 2b?

Answer: The purpose of Fig. 2 is to show that solely from the anisotropy parameters obtained from μ SQUID studies, it is not possible to fully described all QTM events. In Fig. 2b it can be seen the transitions arising from the lowest energy state, and those arising from higher energy states. At 25 mK, and after the system have been fully polarised by applying a -1 T field, the populated electronic state is expected to be the one at the lowest energy (line at the lowest energy in Fig. 2c). Upon sweeping the field from -1 T to $+1$ T, tunnelling can occur just at the crossings with other states when QTM is allowed. This is the case for the transition at 0.048 and 0.067 T (red dotted lines in Fig. 2b). Upon increasing temperatures, higher in energy states become populated, hence, QTM can also become active for crossings between these high-energy states (black dotted lines). As described in the text, the remaining transitions in Fig. 2b cannot be assigned to any crossing in the Zeeman diagram calculated with the parameters obtained from solely μ SQUID results, highlighting the limitations of μ SQUID to determine high-order parameters for complex systems. Figure 2c has been amended to clarify this point. Moreover, in the text we have clarified that at this stage in the MS, it's not possible to assign all steps just from μ SQUID, since collective dynamics, such Spin-Spin Cross-relaxation can also occur, complicating the description of the data. At this stage in the MS, the description and data clearly highlight the need of a more sensitive technique, as the μ SQUID-EPR technique we described.

6. In the next paragraph (pg 4-5), the author mentioned that “Up to this point, the μ SQUID data and the fine structure allows an initial determination of the spin Hamiltonian parameters given by Eq. 4. $k_B T_B \approx 3(|B_2^0|S^2 - |B_2^0|/2)$ allows the estimation of the axial zero-field splitting term leading to $|B_2^0|/2 = 8.5$ mK”. How can the data of Fig. 1, 2 and eq. 4 determine the parameters? And how does the value of 8.5 mK come from?

Answer: The crossover temperature to the superparamagnetic regime, known as blocking temperature (T_B), can be employed for the determination of the anisotropy parameters in SMMs. As described in the MS, through temperature dependent studies, it was found that the loops change from open to totally close loops (superparamagnetic behaviour) at a blocking temperature of $T_B \approx 0.3$ K (Fig. S2). A rough estimation can be obtained by relating the crossover temperature directly to the $|B_2^0|$ parameter by $k_B T_B \approx 3(|B_2^0|S^2 - |B_2^0|/2)$ (for half integer spin) leading to a $|B_2^0| = 8.5$ mK. The structure of μ SQUID loops can likewise be employed for the determination of anisotropy constants in SMMs since the steps in the loops can be related directly to anisotropic parameters (for simple systems) (See for example: *Phys. Rev. Lett.* **2002**, *89*, 197201) through eq. 5, i.e., $H_n = \frac{3nB_2^0}{g\mu_B} \left[1 + \frac{B_4^0}{3B_2^0} ((m-n)^2 + m^2) \right]$. As described in the MS, the Zeeman diagram shown in Fig. 2c was obtained after determining the anisotropic parameters $|B_2^0|$ and $|B_4^0|$ and the crossings obtained from the μ SQUID loops (Fig. 1c,d and 2a,b). Clarification to these points was added to the MS.

7. In the first paragraph of pg 7, the authors mentioned “Figure 3a shows magnetisation curves as a function of frequency obtained while scanning the magnetic field with a fixed sweeping rate of 8 mT/s and by applying radiofrequency pulses of 40 μ s width and a period of 300 μ s in between μ SQUID measurements.” The authors used microwave pulse with 40us and 300us period. Why do the authors choose those time scales? Have you changed to different time scale? Because, for the spin resonance experiment, when you saturate the transition, the absorption signal can mislead for determination of the transition probabilities. Also, I see that you sweep RF in very wide range of high frequencies (up to 21GHz). And in this case, I believe that your microwave line could have very different rate of loss across such a wide range. This means that the RF amplitude over freq sweep can be highly frequency-dependent. How do the authors calibrate this?

Answer: The basic idea of the μ SQUID-EPR technique is to apply RF pulses to the sample in between μ SQUID measurements (see Figure S4).

Figure S4: Timing of the SQUID measurement and the application of RF pulses to the sample.

Thus, the timescale of the EPR should not overlap with the timescale of the μ SQUID technique. For the measurements discussed in the present paper, the phase of the signal is not relevant, just its amplitude and frequency. The pulsed nature of the RF signal is due to the limitations imposed by the μ SQUID technique. It is important to stress that the technique is a quasi-CW EPR experiment, not a pulsed experiment, as the μ SQUID cannot function under RF irradiation, the RF pulses applied to the sample were interleaved with the current pulses used to trigger the μ SQUID (see Fig. S4). The parameters of the RF signal in the experiment were optimized so that the resonant transitions dominate non-resonant ones (heating). We apply RF signals in the [0.1:40] GHz and it is true that the antenna has a very different rate of loss across this range. However, in this work we don't compute (or try to compute) the probability of different transitions, but focus on only identifying the resonant transitions (seen as peaks in the magnetization curve). The coherent manipulation is the goal of the ongoing work on the project and will be the subject of future reports. This description was added to the SI.

8. In the second paragraph of pg 8, the authors mentioned "Most of the transition lines are part of the set with the smallest absolute value of the slope (denoted by 1, 2, 3,..., in Fig. 3b) and correspond to the "allowed" dipolar transitions characterised by the selection rule $\Delta m = \pm 1$. The magnitude of the zero-field splitting obtained from μ SQUID measurements, the linearity and the relative intensity of different lines, indicate that 1 corresponds to the $(7/2, 5/2)$ transition. The subsequent lines result from $(5/2, 3/2)$, $(3/2, 1/2)$, ..., transitions. Moreover, 7, 8, and 9 correspond to higher-order transitions allowed by the nonaxial interactions." what are the transitions for the denoted numbers of 4, 5, 6? If I follow the authors' notation, 4 = $(1/2, -1/2)$, 5 = $(-1/2, -3/2)$, 6 = $(-3/2, -5/2)$. Is this correct? Please explain the transition for 4, 5, 6.

Answer: We thank the referee for the observation. Indeed, the transitions for **4**, **5** and **6** are between $(1/2, -1/2)$, $(-1/2, -3/2)$ and $(-3/2, -5/2)$, as the referee correctly assigned. This has been clarified in the MS.

9. Lastly, I found out many basic typos and grammatic mistakes.

- In line 7, 2nd paragraph of pg 2:
The possibility of chemically modify the periphery ... \diamond The possibility of chemically modifying the periphery ...
- In line 2, after the equation (2) of pg 3:
Spin-orbit admixture in the ground state is responsible ... smaller than in $\langle L \rangle$... \diamond Spin-orbit admixture in the ground state is responsible ... smaller than one in $\langle L \rangle$...
- In line 4, 1st paragraph of pg 5:
0.048 and 0.067mT \diamond 0.048 and 0.067 T
- In line 14, 1st paragraph of pg 7:
for magnitude of the peak Fig. 3. \diamond for magnitude of the peak in Fig. 3.
- In line 6, caption of the Fig. 3 of pg 7:
... Obtained from Eq. 5.7. $(m, m \pm 1)$... \diamond Eq 5, 7

- In line 1, 1st paragraph of pg 10:
Allowing to solve exactly for $B_0(2n)$. The eigenvalues of the axial Hamiltonian as a function of the applied magnetic field are shown in Fig. 3c and ...
◇Allowing to solve exactly for $B_0(2n)$, the eigenvalues ...
- In line 1, 5th paragraph of pg 10:
In Fig. 3c the anticrossings are highlighted... ◇In Fig. 3c, the anticrossings are highlighted...
- In line 4, 5th paragraph of pg 10: corresponds to $\Delta m = 4$ (blue), ... ◇corresponds to $\Delta m = 4$ (blue), ...
- In line 5, 6th paragraph of pg 10: selection rule: $(k, \Delta m)$ Fig. 4c,d. ... ◇selection rule: $(k, \Delta m)$ in Fig. 4c,d. ...
- In line 12, 6th paragraph of pg 10:
hence, in the following discussion $4(4)4(4)$ is the only term that will be considered it incorporates the $4(6)4(6)$ component.
◇I cannot understand this sentence grammatically.

Answer: We thank the referee for the observations and apologise for the misprints. All typos and errors have been revised and amended.

Reviewer 2

The manuscript presents the use of mu-SQUID-EPR for the estimation of the transverse crystal field parameter in SMM. The knowledge of these parameter is very useful in the domain of quantum information processing since a weak variation of bad knowledge of it dramatically change the dynamics property of the system. The results presented in the manuscript is of high interest. The manuscript is clear, well written and made for a broad audience. The author manages to clearly explain very specific concept in spectroscopy which made the manuscript suitable for Nature audience. I have no major concern on it but I have a few suggestions.

1. The manuscript is made for broad audience but on the other hand I think there is a lack of detail for specialist in the field. Although not necessary I think the numeric data of the figure 4 ac and 5 should be provided to allow the reproducibility of the fit process.

Answer: We thank the referee for the comment. The experimental data for figures 4a,c and 5 have been included as supporting information.

2. The fitting and simulation procedure explanations is a bit weak. I suggest the author to add detail (may be in the SI)

Answer: A more detailed explanation regarding the simulation have been added to the SI.

3. The values of B_{kl} are given without error estimation. However, I think the error estimation is very important since some B_{kl} parameters (like B_{44}) are much more sensitive variations of the frequency of resonance than others

Answer: The error bars for the B_{kl} parameters have been included in Tables S2 and S3.

4. Mu-SQUID-EPR is a very specific technic and the author add a recent achievement on Gd3+. <https://doi.org/10.1103/PhysRevApplied.18.014054>

Answer: We thank the referee for the very relevant and important reference. This was included in the MS.

Reviewers' Comments:

Reviewer #1:

Remarks to the Author:

I have found that the authors addressed most of my comments. Based on the reviewer #2's comment, I also have no major concern and agree publication in Nat. Comm.

Reviewer #2:

Remarks to the Author:

The authors answer correctly to all of my remarks.

I consider the manuscript suitable for publication

Response to the Referees

Reviewer 1

I have found that the authors addressed most of my comments. Based on the reviewer #2's comment, I also have no major concern and agree publication in Nat. Comm.

Reviewer 2

The authors answer correctly to all of my remarks. I consider the manuscript suitable for publication.

Thank you very much for considering our manuscript (NCOMMS-22-33063-T) for publication. On behalf of all co-authors, I would like to thank the referees for taking the time to carefully review our manuscript and for their acceptance of the MS.